# Link between Oral Health, Periodontal Disease, Smoking, and Systemic Diseases in Romanian Patients

**DOI:** 10.3390/healthcare11162354

**Published:** 2023-08-21

**Authors:** Christoph Schwarz, Adrian Ioan Hajdu, Ramona Dumitrescu, Ruxandra Sava-Rosianu, Vanessa Bolchis, Diana Anusca, Andreea Hanghicel, Aurora Doris Fratila, Roxana Oancea, Daniela Jumanca, Atena Galuscan, Marius Leretter

**Affiliations:** 1Translational and Experimental Clinical Research Centre in Oral Health, Department of Preventive, Community Dentistry and Oral Health, University of Medicine and Pharmacy “Victor Babes”, 300040 Timisoara, Romania; schwarz.christoph@umft.ro (C.S.); hajdu.adrian@umft.ro (A.I.H.); sava-rosianu.ruxandra@umft.ro (R.S.-R.); bolchisvanessa@gmail.com (V.B.); diana.anusca@ymail.com (D.A.); andreea.hanghicel@yahoo.com (A.H.); roancea@umft.ro (R.O.); jumanca.daniela@umft.ro (D.J.); galuscan.atena@umft.ro (A.G.); 2Department of Preventive, Community Dentistry and Oral Health, University of Medicine and Pharmacy “Victor Babes”, Eftimie Murgu Sq. No. 2, 300041 Timisoara, Romania; 3Faculty of Dental Medicine, Ludwig-Maximilian-University Munich, Goethestraße 70, 80336 München, Germany; a.fratila@campus.lmu.de; 4Department of Prosthodontics, Multidisciplinary Center for Research, Evaluation, Diagnosis and Therapies in Oral Medicine, University of Medicine and Pharmacy Timisoara “Victor Babes”, 2 Eftimie Murgu Square, 300041 Timisoara, Romania; mariusleretter@yahoo.com

**Keywords:** oral health, periodontal status, smoking, systemic diseases

## Abstract

(1) Background: The link between oral and systemic health is becoming increasingly obvious. Oral diseases, particularly periodontitis, have been linked to various diseases including diabetes and cardiovascular disease, among others. This survey aimed to assess the oral health condition of individuals, considering both their overall health and periodontal status, by performing oral examinations and collecting data using questionnaires. (2) Methods: After obtaining approval from the University’s Ethics Committee, the study was carried out from 2021 to 2022 at the Department of Oral Health, located in the Emergency Municipal Hospital in Timisoara, Timis County, Romania. Bivariate correlations were performed using nonparametric Spearman’s Rho using SPPS software version 23. To assess the importance of smoking frequency related to the severity of periodontitis diagnosis, the ANOVA Simple test (one-way) and Hochberg GT2 post hoc analysis were utilized. The chi-squared test was employed for nominal variables. A significance level of 0.05 (alpha = 0.05) was adopted for all statistical tests. (3) Results: There is a significant positive association between the frequency of systemic disease and the severity of the periodontitis diagnosis taken as a total, Rho (242) = 0.151, *p* < 0.05, and taken as a stage, Rho (242) = 0.199, *p* < 0.01, thus as the severity of the diagnosis increases, the patient presents comorbidities. Hochberg GT2 post hoc analysis indicates that the non-smoking group has statistically significantly lower diagnostic severity (Mdif = −0.81, *p* = 0.01), with a strong effect size (Cohen’s d = 0.73). (4) Conclusions: The findings are increasingly indicating a potential association between oral diseases and a range of systemic diseases. The impact of periodontal disease on the quality of life is significant, especially in individuals with associated systemic conditions and present risk factors.

## 1. Introduction

A healthy mouth is intricately linked with overall health and the quality of one’s life. In fact, it can rightfully be considered a fundamental human entitlement [1,2]. The prevalence of oral disorders, as a group, represents the most prevalent chronic illnesses experienced by humanity, significantly affecting crucial oral functions, self-confidence, quality of life, and overall health and wellness [3,4]. Over 3.5 billion individuals are affected by oral diseases, with untreated dental caries in permanent teeth being the most prevalent. Severe periodontal disease affects nearly 10% of the world’s population, while dental caries in primary teeth affect over 530 million children. Oral diseases have a higher impact on impoverished and socially disadvantaged communities. Furthermore, many oral diseases are associated with other noncommunicable conditions such as cardiovascular diseases, diabetes, cancers, pneumonia, obesity, and premature birth [5].

Systemic diseases can directly affect oral health through pathological mechanisms or indirectly through changes in behavior resulting from the disease or its treatment. Similarly, alterations in oral health can have repercussions on systemic well-being [6]. The oral microbiome contains a diverse range of bacteria, with around 700 different phylotypes and approximately 400 species found in subgingival plaque [7,8,9]. In periodontitis, the subgingival microflora can harbor numerous bacterial species, but only a small number are considered etiologically important in the progression of the disease [10,11,12]. The prevalence, extent, and severity of oral health tend to increase with age. Some studies suggest that the rate of periodontal destruction remains constant until the age of 70, indicating that age is not a significant risk factor for individuals under 70. However, with increased life expectancy, there is a growing geriatric population that will require oral health treatment, particularly for periodontal diseases [9]. Smoking has been consistently linked to periodontal disease and tooth loss. The association between smoking and periodontal disease implies that smoking is likely a major contributor to tooth loss. The connection between tobacco smoking and acute necrotizing ulcerative gingivitis was first reported in 1947 [13].

Early and adequate diagnosis and treatment are essential to reduce the burden of periodontitis. Moreover, it is important to identify the most severe cases of the disease, as they are the most challenging therapeutically, with poorer tooth prognosis, a higher risk of tooth loss, additional costs, and a greater likelihood of influencing systemic status and individual health [14]. Therefore, periodontitis represents a major public health problem due to its high prevalence and negative impacts affecting oral health (dental losses and disabilities, aesthetic, and masticatory dysfunction) as well as overall health and quality of life (systemic impacts, malnutrition) with significant psychosocial and economic repercussions [15,16].

In 2021, the World Health Organization (WHO) granted approval for a Resolution concerning oral health. The World Health Assembly Resolution paves the way for better oral health care. Rather than relying solely on conventional curative methods, the WHO emphasizes the crucial role of prevention and the inclusion of oral health in universal health coverage programs [5]. 

In Romania, there is limited epidemiological data on oral health status and in particular, regarding periodontitis. To our knowledge, there has been no organized effort to screen various population groups in Romania systematically or even as part of a planned initiative. Therefore, this study aimed to evaluate the oral health of adult individuals in Romania who have periodontal disease and assess the relevance of the connections between the stage and grade of periodontal disease with systemic conditions, according to the New Classification System for Periodontal and Peri-implant Diseases and Conditions (WWC2017), attending the Department of Oral Health at the Emergency Municipal Hospital in Timisoara, Timis County, Romania.

## 2. Materials and Methods

The study was conducted between 2021 and 2022 on patients attending the Department of Oral Health in Timisoara, Timis County, Romania. The study protocol was reviewed and approved by the Ethics Committee of the University (Approval Number 34/2018) and all the participants in the study gave their written, informed consent.

### 2.1. Study Group

Patients from the Department of Oral Health in Timisoara, Timis County, Romania, were included to evaluate the oral-health status in individuals, taking into account their general state of health and periodontal status. Following the application of the exclusion criteria, 21 patients were excluded from the initial cohort of 263, leaving a final sample of 242 patients for analysis. Subsequently, a consecutive group of 242 patients seeking oral evaluation at the Department of Oral Health underwent examination as part of the study. All patients completed a self-reporting questionnaire assessing information about demographical data, general health problems, and smoking habits.

Inclusion criteria: patients with and without systemic diseases and patients being able to and willing to give their written and informed consent.

Exclusion criteria: patients who refused to complete the questionnaires and who did not present themselves to the oral-health evaluation.

### 2.2. Oral Examination

All patients included in the study were examined by resident dentists with a general dentistry specialization from Timisoara Clinical Municipal Emergency Hospital during their training stage at the Department of Oral Health, Faculty of Dental Medicine, University of Medicine and Pharmacy “Victor Babes” Timisoara. Each patient in the study group underwent a comprehensive full-mouth periodontal examination, and a periodontal chart was meticulously filled out for every individual. To ensure consistency and accuracy, the investigators (V.B., D.A., A.H.) conducting the examinations underwent prior calibration. Before commencing the study, they received detailed written instructions on the study’s design, periodontal evaluation, and data collection protocols. Additionally, they participated in two training meetings, overseen by a senior dentist (A.G.), to further enhance their expertise and standardize the examination process. The dentists employed a plane examination mirror and a probe (1 mm marking periodontal probe—UNC-15 periodontal probe, Hu-Friedy, Chicago, IL, USA) to assess the presence and extent of periodontal disease while documenting additional oral health issues. To maintain intra-examiner reliability, they utilized precise instruments, adhered to recommended oral examination methods, and diligently recorded the findings for every patient. The local risk factors present, the average probing depth, the average level of gingival attachment, the percentage of bacterial plaque, the percentage of bleeding on probing, and the stage and grade of the periodontal diagnosis were recorded in the chart of each patient.

### 2.3. Population Characteristics

To gather further details about the patients participating in the study, the investigators administered a self-reporting questionnaire that documented the following information:-Demographic data (age, sex, and social background—urban or rural);-Smoking habits (non-smoker, former smoker, or active smoker with tobacco exposure noted as number of cigarettes smoked per day);-General state of health;-Use or non-use of different types of medication.

### 2.4. Data Analysis

Univariate descriptive statistics (mean, standard deviation, and standard error) were used to evaluate the sample’s characteristics.

A cross-sectional single-center correlational study on a total of 242 patients was performed. The data were analyzed using SPSS statistical software. Nonparametric Spearman Rho bivariate correlations were used to see if there was a relationship between the following variables: totals from the periodontal sheet, periodontal sheet items taken separately, mean age of patients, frequency of systemic conditions, total diagnosis, diagnosis criteria taken separately, and gender of patients. 

The ANOVA Simple test (one-way) was used to compare the groups of smokers by frequency level and non-smokers or ex-smokers by degree and stage at diagnosis and periodontal chart. For post hoc analysis, it was taken into account that there is an uneven number of participants in the groups and that the data dispersion is not homogeneous. Therefore, the Hochberg GT2 post hoc procedure specifically designed for such situations was utilized.

For nominal variables, the chi-squared test was applied. A threshold value for alpha of 0.05 was used for all the statistical tests.

## 3. Results

The group of participants consisted of 242 patients (109 males and 133 females) attending the Department of Oral Health Timisoara. The mean age was M = 44.5, (±12.19), as shown in Table 1. 

In terms of smoking habits, of the 242 patients examined, 125 of them declared that they were non-smokers, 13 were former smokers, 32 of them smoked less than 10 cigarettes a day, 40 of them smoked between 10 and 20 cigarettes a day, and 32 of them smoked more than 20 cigarettes a day (see Table 2).

### 3.1. Correlations between Age, Gender, Presence of Bacterial Plaque, Gingival Bleeding, and Systemic Diseases

Regarding the age of the patients, the Spearman Rho correlation shows that there is a significant positive relationship between the mean age and the mean value of the probing depth (an item in the periodontal chart), Rho (238) = 0.18, *p* < 0.01. The same is true for total diagnosis, with a significant positive association between age and total at diagnosis, Rho (238) = 0.36, *p* < 0.001, with a significant positive association between age and degree of diagnosis. Rho (238) = 0.18, *p*< 0.01, and a significantly positive association between age and stage of diagnosis, Rho (238) = 0.46, *p* < 0.001. As for age and somatic conditions, again a significant positive association can be observed between the two variables, Rho (238) = 0.44, *p* < 0.001. It can be observed that the older the patient, the greater the depth of probing, the greater the severity of the diagnosis of periodontitis, both overall and taken by grade and stage, and the greater the frequency of somatic complaints (see Table 3). 

There is a significant positive association between the frequency of systemic disease and the severity of the diagnosis of periodontitis taken as a total, Rho (242) = 0.15, *p* < 0.05, and taken by stage, Rho (242) = 0.19, *p* < 0.01. Thus, we can observe that as the severity of the periodontitis increases, the patient presents comorbidities. 

### 3.2. Correlation between Oral Health and the Presence of Systemic Diseases

Out of the group of 242 patients, 165 of them declared that they do not have systemic issues, 24 patients have been diagnosed with either heart problems or high blood pressure, 3 of them have diabetes, 3 have respiratory diseases, and 2 have blood clotting issues. As shown in Figure 1, there is a correlation between the presence of systemic conditions, especially cardiovascular diseases, and the presence of local factors, namely dental plaque, calculus, dental crowding, and overhanging restorations. A total of 50.88% of patients with local risk factors such as calculus and bacterial plaque have systemic conditions like cardiovascular diseases and hypertension, diabetes, and metabolic syndrome. Among patients with bacterial plaque, extensive restorations, and traumatic occlusion, a percentage of 7.02% have hypertension and cardiovascular pathology. Bacterial plaque, calculus, and traumatic occlusion are present in 14.04% of patients with declared respiratory diseases. 

### 3.3. Correlation between Smoking Habits and Oral Health Status

Patients were divided into five groups: one non-smoker (N = 125), one ex-smoker (N = 13), one smoking less than 10 cigarettes per day (N = 32), one smoking between 10 and 20 cigarettes per day (N = 40), and one smoking more than 20 cigarettes per day (N = 32). The results indicate that there are significant differences between the groups compared, F (4, 237) = 6.64, *p* < 0.001. The lowest mean diagnostic score was in the group smoking less than 10 cigarettes per day (M = 5.06, SD = 1.50), followed by the group of non-smokers (M = 5.74, SD = 1.26), then the group of ex-smokers (M = 5.9, SD = 1.03), then the group of smokers who smoke between 10 and 20 cigarettes per day (M = 6.2, SD = 1.34), and the last being the group of smokers who smoke more than 20 cigarettes per day (M = 6.56, SD = 0.94). The Hochberg GT2 post hoc analysis indicates that the non-smoking group has a statistically significantly lower diagnostic severity (Mdif = −0.81, *p* = 0.01), with a strong effect size (Cohen’s d = 0.73). The group smoking less than 10 cigarettes per day also has significantly lower diagnostic severity of periodontitis than the group smoking 10–20 cigarettes per day (Mdif = −1.13, *p* < 0.01), with a strong effect size (Cohen’s d = 0.79). The group smoking less than 10 cigarettes per day also has a statistically significantly lower level of diagnostic severity than the group smoking more than 20 cigarettes per day (Mdif = −1.50, *p* < 0.001), with a very strong effect size (Cohen’s d = 1.10). For the rest of the comparisons, no statistically significant differences were observed. Thus, on the basis of the simple ANOVA analysis, we can consider that non-smokers have less severity by degree and stage of diagnosis than those who smoke more than 20 cigarettes per day. Those who smoke less than 10 cigarettes per day have lower severity compared to other smokers, but not to non-smokers (Table 4).

## 4. Discussion

Our study confirms the link between oral health, periodontal disease, smoking, and systemic diseases. Apart from bacterial plaque, several other factors can influence the host response and increase an individual’s susceptibility to periodontal disease. These factors include age, systemic conditions, and tobacco use. These elements play a significant role in modifying the oral environment and can contribute to a heightened risk of developing periodontal issues. Population surveys in both developed and developing countries have extensively measured the prevalence and severity of periodontal disease. These studies have been conducted with diverse objectives, designs, and measurement criteria, providing valuable insights into the global burden and variations of periodontal conditions across different regions.

The significance of oral health in today’s world extends far beyond mere dental hygiene. It now plays a pivotal role in determining one’s overall quality of life. Just like general health, the intricate connections between oral health and well-being involve intricate mechanisms that are multifaceted and influenced by personal beliefs and subjective values [12]. Oral health extends beyond its traditional focus on teeth alone and encompasses a broader perspective, including oral epidemiology. It is important to recognize that systemic conditions are significantly influenced by oral health. The oral cavity is considered the “window to general health” and one cannot stay healthy without good oral health [12,13].

The interconnections between oral health, overall health, and well-being are vital and should not be overlooked or dismissed. Adopting health-promoting behaviors, such as receiving oral hygiene instruction or enhancing oral health literacy, can greatly improve the majority of oral diseases. Furthermore, it is worth noting that oral diseases and the top four noncommunicable diseases—cardiovascular disease, cancer, chronic respiratory disease, and diabetes—share common risk factors. A risk factor refers to an element such as environmental exposure, individual behavior, lifestyle choices, or inherited characteristics that are linked to and could potentially lead to the host’s exposure to a disease [14]. The concept of shared risk underscores the idea that the factors contributing to an individual’s susceptibility to periodontitis can also make them vulnerable to other systemic diseases [15].

These risk factors include cigarette smoking, alcohol consumption, and sugar intake. Our current findings are in accordance with other studies that revealed the fact that tobacco smoking, an environmental risk factor associated with periodontitis, is also linked to systemic issues like cardiovascular and respiratory diseases, as well as diabetes. Other factors, such as age, stress, and male gender, also contribute to these health problems [16,17,18]. A recent study including data from 17 European countries revealed an overall smoking prevalence estimate of 13.4% among Europeans aged 65–74 years, which was significantly higher than for people aged ≥75 years (8.2%) [19]. By modifying these shared risk factors, we can efficiently prevent both oral diseases and potentially life-threatening systemic conditions. Implementing preventive strategies effectively slows down these well-known risks [20]. Although the average lifespan in most western societies is increasing, both systemic and oral health deteriorates with aging, and can reduce quality of life, even while life span expands thanks to modern lifestyle [21,22]. Hence, it is crucial to carefully analyze and identify various risk factors, such as age, gender, smoking habits, and medical conditions, that can potentially influence oral health and treatment requirements. 

The information available in the current study provided a more accurate assessment of the link between periodontal disease and smoking. By categorizing individuals as current, former, and non-smokers, potential biases were minimized, leading to a more robust analysis of the evidence. This approach allowed for a more precise evaluation of the association between periodontal disease and cigarette smoking. Our current findings support the conclusions of other studies, indicating that smoking significantly increases the risk of developing and worsening periodontal diseases [23,24,25,26]. Numerous cross-sectional studies have revealed that smokers have a notably higher likelihood, ranging from two to seven times, of experiencing periodontitis compared to nonsmokers [22,23,27]. Furthermore, smoking has been linked to tooth loss in individuals undergoing periodontal maintenance [23]. This evidence underscores the critical role of smoking cessation in promoting better periodontal health and overall well-being. According to reports, smokers tend to exhibit a notably higher plaque index, and interestingly, they have a smaller average number of bleeding sites (27%) compared to nonsmokers (40%). This suggests that smoking might have complex effects on oral health, influencing both plaque formation and bleeding tendencies in the gums [24]. As a crucial component of the healthcare system, dental professionals ought to educate patients about the advantages of quitting smoking.

Our findings showing a positive significant correlation between age and probing depth are in accordance with other studies, which demonstrate that the overall prevalence of periodontitis increases with age, and the incidence rises steeply in adults aged 30–40 years. Such burden of periodontitis will continue to increase with the growing ageing population and also due to increased tooth retention globally [17]. Recent findings indicate that there is early evidence supporting the notion that periodontitis, a condition characterized by the inflammation of the gums and loss of support around the teeth, becomes more prevalent and severe with advancing age. This suggests that age can serve as a potential indicator for the deterioration of periodontal health and subsequent tooth loss [28,29]. Albandar [30] observed a correlation between age and worsening periodontal condition among North American patients, with disease severity playing a significant role in this relationship. Furthermore, it is worth noting that the decline in periodontal health may be primarily attributed to tooth loss, with the older age group experiencing the most severe manifestations of the disease. 

There is a limited number of published studies exploring the impact of periodontal disease on the quality of life in patients with chronic conditions such as diabetes, or cardiovascular disease [31]. In recent years, there has been a growing focus on exploring potential connections between periodontal disease and systemic diseases. Determining the exact role that systemic diseases may play in the development of periodontitis is challenging and it is essential to carefully match control groups in terms of age, gender, oral hygiene, and socioeconomic status [32]. This has generated significant interest among the general public, who are becoming increasingly conscious of the possible existence of these links and, in some cases, expressing concerns about the implications for their own health. It is noteworthy that almost half of the adult population in the United Kingdom experiences some form of periodontal disease, leading to a substantial number of individuals seeking dental care on a daily basis [33]. In Europe, detailed elaborate periodontal epidemiologic studies have only been conducted in Scandinavia (Halling & Björn 1987, Papanaou et al., 1988, 1989, 1991, Papanaou & Wennström 1990, Wennstro¨m et al., 1993, Serino et al., 1994, Eliasson & Bergström 1997, Papanaou 1999, Skudutyte-Rysstad et al., 2007, Hugoson & Norderyd 2008, Hugoson et al., 2008) and in certain Western European countries (Schürch et al., 1988, Diamanti- Kipioti et al., 1995) [23].

Borrell and Papapanou [34] collected periodontal data to assess the effect of risk factors, and suggested that the most prominent are age, sex, smoking status, educational and socioeconomic status, and diabetes.

Furthermore, understanding the connection between oral health and systemic health among patients can profoundly influence their access to oral care. The results of our study support the existence of a connection between periodontal disease and oral health, as well as systemic diseases. A significant proportion of patients (two-thirds) suffering from chronic periodontal disease were unaware of the connection between periodontal therapy and systemic diseases [30]. This lack of awareness highlights the importance of educating patients about the potential impacts of periodontal treatment on their overall health.

To our knowledge, in Romania, there is a scarcity of studies that examine the impact of periodontal disease on the quality of life specifically in restricted patients [35,36]. These studies typically involve small or specific population samples. The research among young individuals who sought treatment at a prosthodontics department in a university located in Western Romania revealed a significant occurrence of periodontal disease [37]. Our study results demonstrated that 50.88% of the investigated patients with periodontal disease have cardiovascular conditions. This is in accordance with another Romanian study that was conducted between 2018 and 2019 to investigate the prevalence of periodontitis among patients with associated cardiovascular disease, a condition that is more common in this population compared to other European [38]. Cardiovascular disease is a significant cause of mortality in Europe, with recent statistics showing that 20% of deaths are attributed to ischemic heart disease. Given its prevalence and impact, exploring the link between periodontitis and systemic disease is crucial in this context. While the elevated chance of cardiovascular disease linked with periodontal disease might appear moderate (approximately 20%), even this slight rise could carry significant implications for public health regarding cardiovascular disease and strokes. This is due to the widespread prevalence of periodontal disease in the population [39].

Our study has provided useful evidence that can be utilized to improve the knowledge and understanding of the dental community to enhance oral seeking behaviors of patients. Nevertheless, there were certain limitations. The present data may not truly represent a large community of patients. Nonetheless, the established positive correlations could hold significance, even in light of the relatively limited size of the studied population group. Also, one notable limitation is the method of categorizing patients’ smoking status, which was solely based on self-reported data from their general medical history. This resulted in three categories: current smokers, former smokers (self-reported ex-smokers), and non-smokers (those who reported never smoking). However, we acknowledge that a more comprehensive tobacco use history, encompassing relevant details such as daily consumption and time since cessation, was not recorded within our sample. This absence of detailed information on smoking habits could potentially limit the depth of our analysis and interpretation of the study findings.

Therefore, the management and prevention of these systemic conditions necessitate effective control of oral diseases. Promoting primary prevention of periodontal diseases could yield advantages extending beyond dental well-being, with potentially significant implications for overall health. A future study should try to discover whether the provision of knowledge about the oral-systemic link actually increased the number of patients seeking oral care, increased the number of routine dental visits, or increased the number of dental procedures provided to the patients.

## 5. Conclusions

In conclusion, smoking and systemic diseases are correlated with periodontal disease, emphasizing the need for a strong interdisciplinary collaboration. By addressing these factors and fostering cooperation among healthcare professionals, the management and prevention of periodontal disease could improve, ultimately enhancing the overall oral and systemic health of patients. To shed more light on this matter, larger and long-term randomized intervention trials are necessary. These findings carry important implications for assessing, planning, and treating periodontal disease, as well as evaluating the effectiveness of periodontal care.

## Figures and Tables

**Figure 1 healthcare-11-02354-f001:**
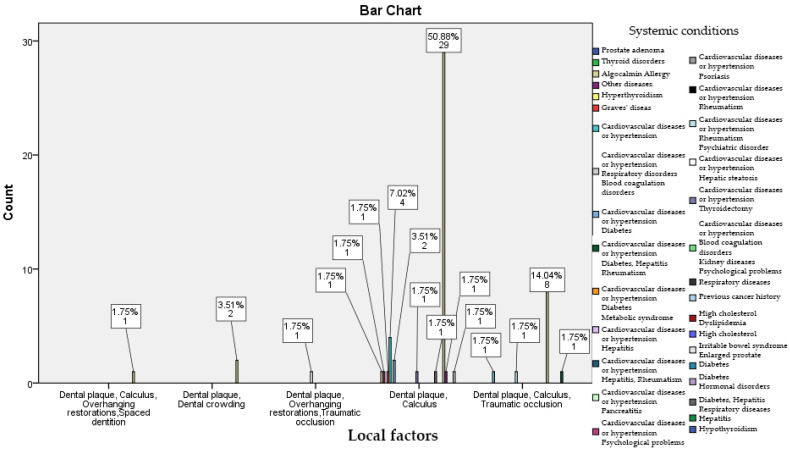
Correlation between oral health and the presence of systemic diseases.

**Table 1 healthcare-11-02354-t001:** Sample description according to gender and place of residence.

Place of Residence	Gender	
	Male	Female
**Urban**	89 (44.1%)	113 (55.9%)
**Rural**	24 (60%)	16 (40%)

**Table 2 healthcare-11-02354-t002:** Sample description according to smoking habits.

Smoking Habits	N
Non-smoker	125
Former smoker	13
Smoker < 10 cigarettes/day	32
Smoker 10–20 cigarettes/day	40
Smoker > 20 cigarettes/day	32
Total	242

**Table 3 healthcare-11-02354-t003:** Correlations between age, gender, presence of bacterial plaque, gingival bleeding, and systemic diseases.

Personal Parameters	Presence of Bacterial Plaque	Gingival Bleeding	The Average Value of Sounding Depth	Stage of Diagnosis	Level of Diagnosis	Total Score Diagnostic	Systemic Diseases
**Age**	−0.55	−0.10	0.18 **	0.46 **	0.18	0.36 **	0.44
	0.39	0.10	0.00	0.00	0.00	0.00	0.00
	237	237	238	238	238	238	238
**Gender**	−0.01	−0.06	−0.03	−0.02	−0.05	−0.06	0.05
	0.78	0.30	0.54	0.63	0.40	0.33	0.41
	240	240	242	242	242	242	242

** *p* < 0.01.

**Table 4 healthcare-11-02354-t004:** Correlation between smoking habits and the oral health status—post hoc tests (Hochberg).

	Non-Smoker (Mean, SD, CI 95%)	Former Smoker (Mean, SD, CI 95%)	Smoker < 10 Cigarettes/Day (Mean, SD, CI 95%)	Smoker 10–20 Cigarettes/Day (Mean, SD, CI 95%)	Smoker > 20 Cigarettes/Day (Mean, SD, CI 95%)
Non-smoker	-	−0.17 (0.36, −1.21–0.86)	0.68 (0.25, −0.02–1.38)	−0.045 (0.22, −1.10–0.19)	−0.81 * (0.25, −1.15–1.11)
Former smoker	0.17 (0.36, −0.86–1.21)	-	0.86 (0.41, −0.31–2.03)	−0.27 (0.40, −1.41–0.86)	−0.63 (0.41, −1.81–0.53)
Smoker < 10 cigarettes/day	−0.68 (0.25, −1.38–0.26)	−0.86 (0.41, −2.03–0.31)	-	−1.13 * (0.29, −1.98–−0.29)	−1.15 * (0.31, −2.39–−0.60)
Smoker 10–20 cigarettes/day	0.45 (0.22, −0.19–1.10)	0.27 (0.40, −0.86–1.41)	1.13 (0.29, 0.29–1.98)	-	−0.36 (0.29, −1.20–0.48)
Smoker > 20 cigarettes/day	0.81 (0.25, 0.11–1.52)	0.63 (0.41, −0.53–1.81)	1.15 * (0.31, 0.60–2.39)	0.36 (0.29, −0.48−1.20)	-

* *p* < 0.05.

## Data Availability

The data presented in this study are available on request from the corresponding author. The data are not publicly available in accordance with consent provided by participants on the use of confidential data.

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
