# Peer review of "Link between Oral Health, Periodontal Disease, Smoking, and Systemic Diseases in Romanian Patients"

_healthcare, 2023, doi:10.3390/healthcare11162354_

Round 1

Reviewer 1 Report

Dear Authors,

Manuscript title: Exploring the Link Between Oral Health, Periodontal Disease,  Smoking, and Systemic Diseases: Unveiling the Hidden Connections, is a qualitative approach towards finding a link between oral health and general health. there are few modifications required

Abstract: Well-written and simple. keywords: simple and understandable.

Introduction: Aim of study is not concise. Secondly, elaborated introduction, no need to add findings that are irrelevant. 

Material and method: What was the criteria for periodontal examination? How did one dentist perform all the examination in rural as well as urban area? How was the sample size determined? 

Results: Well-written and tables and figures are properly marked 

Discussion: Authors have not supported few findings of results in discussion. check and correct. However, clinical significance is missing. 

Conclusion: well-written and supports the aim and result of the study. 

References are properly marked and no duplication is found. 

check spellings and grammar. minor corrections are required . 

Author Response

Dear Authors,

Manuscript title: Exploring the Link Between Oral Health, Periodontal Disease, Smoking, and Systemic Diseases: Unveiling the Hidden Connections, is a qualitative approach towards finding a link between oral health and general health. There are few modifications required.

 Response:

Dear Reviewer,

Thank you for taking the time to review our manuscript titled "Exploring the Link Between Oral Health, Periodontal Disease, Smoking, and Systemic Diseases: Unveiling the Hidden Connections." We appreciate your valuable feedback and have carefully considered your comments. Based on your suggestions, we have made the necessary modifications to improve the quality of our article. All the changes have been made within the body of the document and highlighted. Below is a detailed response addressing each of your points:

Abstract: Well-written and simple. keywords: simple and understandable.

Response: Abstract.  We are glad you found the abstract well-written and simple. We have made sure to maintain its clarity and readability throughout the manuscript.

Introduction: Aim of study is not concise. Secondly, elaborated introduction, no need to add findings that are irrelevant.

 Response: Introduction. We apologize for the lack of conciseness in the aim of the. We have revised the aim to make it more precise and relevant: “Therefore, this study aimed to evaluate the oral health of adult individuals in Romania who have periodontal disease and assess the relevance of the connections between the stage and grade of periodontal disease with systemic conditions, according to the New Classification System for Periodontal and Peri-implant Diseases and Conditions (WWC2017) attending the Department of Oral Health at the Emergency Municipal Hospital in Timisoara, Timis County, Romania.”

Material and method: What was the criteria for periodontal examination? How did one dentist perform all the examination in rural as well as urban area? How was the sample size determined? 

Response: Materials and Methods. We understand your concerns regarding the criteria for periodontal examination and the determination of the sample size. In the revised manuscript, we have included a detailed explanation of the criteria used for periodontal examination, the training and calibration process of the dentist and the rationale behind the chosen sample size.

Regarding the examinations, they were conducted only in the urban environment, within the Department of Oral Health. However, the recorded demographic data indicate that the patients come from both rural and urban areas.

Results: Well-written and tables and figures are properly marked 

Response: Results. We appreciate your positive feedback on the presentation of results and the proper labeling of tables and figures. We have ensured the accuracy and clarity of the results section.

Discussion: Authors have not supported few findings of results in discussion. check and correct. However, clinical significance is missing.

Response: Discussion. We acknowledge your observation that some findings from the results were not adequately supported in the discussion section. In the revised manuscript, we have reevaluated and provided a more comprehensive discussion of all the results, supporting them with relevant literature and explanations.

Conclusion: well-written and supports the aim and result of the study. 

Response: Conclusion. We are pleased that you found the conclusion well-written and supportive of the aim and results of the study. We have kept the conclusion intact and made sure it remains consistent with the revised manuscript.

References are properly marked and no duplication is found. 

 Response: References. Thank you for confirming the accuracy of our references. We have meticulously reviewed the reference list to maintain its correctness and avoid any duplication.

Once again, we would like to express our gratitude for your valuable feedback, which has undoubtedly helped us improve the quality and clarity of our article. We believe that the revised manuscript now addresses all your concerns adequately. We hope that you find the updated version more suitable and look forward to your evaluation.

Thank you for your time and consideration.

Sincerely,

Ramona Dumitrescu

Translational and Experimental Clinical Research Centre in Oral Health, Department of Preventive, Community Dentistry and Oral Health, University of Medicine and Pharmacy “Victor Babes”, 300040 Timisoara, Romania

[email protected]  Tel: +40745505481

Also, please see the attachment

Reviewer 2 Report

Thank you for the possibility to review this crosssectional study on the influence of graded smoking on periodontal disease. The correlation to other periodontal risc factors could be presented mor in detail. Please see enclosed my comments and suggestions to improve the paper.

The figure could be improved with a better more extensive and detailed presentation of the large number of bars. What means counts?

Author Response

Thank you for the possibility to review this crosssectional study on the influence of graded smoking on periodontal disease. The correlation to other periodontal risk factors could be presented mor in detail. Please see enclosed my comments and suggestions to improve the paper.

Response:

Dear Reviewer,

Thank you for taking the time to review our cross-sectional study on the influence of graded smoking on periodontal disease. We appreciate your valuable feedback and have carefully considered your comments and suggestions to enhance the paper.

Correlation to other periodontal risk factors: We acknowledge the importance of providing a more detailed presentation of the correlation between graded smoking and other periodontal risk factors. In our revised manuscript, we included an expanded discussion on this aspect, incorporating additional data and analyses to elucidate the relationships to other various risk factors for periodontal disease. This will strengthen the overall findings and contribute to a more comprehensive understanding of the subject matter.

The figure could be improved with a better more extensive and detailed presentation of the large number of bars. What means counts?

Response:

Figure improvement: We understand your concern about the figure's clarity and presentation. Initially we used a figure generated by the statistics program, but we revised it to make it easier to understand. To address this, we revised the figure to make it clearer, more understandable, and reader-friendly. We considered using a different visualization method, to present the large number of bars in a more organized and easily interpretable manner. Additionally, we included a clearer legend that explains what the counts represent, ensuring that readers can easily comprehend the information conveyed by the figure.

We carefully followed the suggestions and comments in the document and have excluded the sentences as you suggested for removal. All the changes have been made within the body of the document and highlighted.

  • Second part of the sentence is confusing.

Response: In regard to the second part of the sentence that caused confusion, it has been rephrased as follows: “The impact of periodontal disease on the quality of life is significant, especially in individuals with associated systemic conditions and present risk factors. “

  • Additionally, we have inserted the citations where requested (see citations 7-9, 10-12).

  • Enclose number and date of the approval

Response: The ethics approval number has been included in the text: “Approval Number 34/2018”.

  • Dental or which type of periodontal probe?

Response: The periodontal probe was used:”1 mm marking periodontal probe—UNC-15 periodontal probe, Hu-Friedy, Chicago, IL, USA”

  • What means precise? pressure calibrated probes?

Response: The term "precise instruments" refers to tools or devices that are specifically designed to provide accurate and detailed measurements or observations. In the context of the sentence provided, it means that the examiners used instruments (periodontal probe) that were finely calibrated and capable of delivering precise results during their oral examinations. These instruments are essential to maintain reliability and consistency in the examination process.

  • Calculation of the number of patients needed was performed?

Response: The initial cohort consisted of 263 patients who sought consultation and various oral health issues at the Department of Oral Health between September 2021 and March 2022. Among them, 21 were excluded based on the exclusion criteria as they declined to provide consent for study participation, resulting in a final sample of 242 participants.

  • Please discuss why you differentiated between patients living in rural county and urban city.

Response: We recorded demographic data related to the patients' backgrounds to track the access to oral health services based on their origin (rural or urban). In the urban environment, there is also greater accessibility to dental services compared to the rural setting.

  • Counts means number of patients? What mean the bars without text? I emphasize another form of figure. These data could be discussed with other studies.

Response: Yes, counts mean number of patients. We revised the figure to make it clearer, more understandable, and reader-friendly.

  • This part (223-285) with cited literature belongs in a shorter form more to the introduction as the results of the cross-sectional study are not discussed.

Response: We reformulated the discussion section, introducing new paragraphs. All the changes can be seen highlighted in the text.

Once again, we sincerely appreciate your constructive feedback, as it will undoubtedly enhance the quality and impact of our research. We are committed to make the necessary improvements to the manuscript based on your suggestions, as your input is highly valuable to us.

Thank you for your support and consideration.

Sincerely,

Ramona Dumitrescu

Translational and Experimental Clinical Research Centre in Oral Health, Department of Preventive, Community Dentistry and Oral Health, University of Medicine and Pharmacy “Victor Babes”, 300040 Timisoara, Romania

[email protected]   Tel: +40745505481

Also, please see attachment

Reviewer 3 Report

The topic of links between oral health, periodontitis, smoking and systemic diseases are important. I have some suggestions to improve the manuscript.

Line 31: SPSS (and not SPPS)

Line 32: "Severity of diagnosis". In fact, I am confused to what "severity" means in your manuscript. Do you mean severity of oral health in general, severity of periodontitis, or severity of the systematic diseases? All of these are diseases and they all have degrees of severity. Please clarify which disease severity you are referring to in your Abstract and in the entire manuscript, as this significantly slowed the flow of the paper and is confusing to the reader.

Line 194: "As shown"

Discussion:

This portion requires a lot of work! 

The paragraph starting at Line 243 is too long and needs to be broken up into at least 3 paragraphs. 

There are a lot of citations of other people's work, but there are few discussions to how YOUR findings support these references. Perhaps more "Our current findings support ..."

There are also many general statements to why oral health and systemic health is important, but very little discussion on why your research is contributing to solving the problems. 

Line 47: "remarkable and invaluable treasure" is odd

Please clarify which "disease" you are referring to (systemic diseases, oral diseases, periodontal diseases)

Author Response

The topic of links between oral health, periodontitis, smoking and systemic diseases are important. I have some suggestions to improve the manuscript.

Response:

 Dear Reviewer,

Thank you for your valuable feedback on our manuscript regarding the links between oral health, periodontitis, smoking, and systemic diseases. We appreciate the time and effort you have put into reviewing our work. We have carefully considered your suggestions and have made the necessary revisions to improve the clarity and overall quality of the article. All the changes have been made within the body of the document and highlighted.

Below is a point-by-point response to your comments:

Line 31: SPSS (and not SPPS)

Response: Line 31: We apologize for the typographical error. It should indeed be "SPSS" and not "SPPS," and we have corrected it accordingly.

Line 32: "Severity of diagnosis". In fact, I am confused to what "severity" means in your manuscript. Do you mean severity of oral health in general, severity of periodontitis, or severity of the systematic diseases? All of these are diseases and they all have degrees of severity. Please clarify which disease severity you are referring to in your Abstract and in the entire manuscript, as this significantly slowed the flow of the paper and is confusing to the reader.

Response: Line 32: We apologize for the confusion regarding the term "severity" in our manuscript. To clarify, in our study, "severity" refers specifically to the severity of periodontitis. We will make sure to mention this explicitly in the Abstract and throughout the manuscript to avoid any confusion for the readers.

Line 194: "As shown"

Response: Line 194: We have corrected the word in the manuscript, and we apologize for the editing mistake.

Discussion:

This portion requires a lot of work! 

The paragraph starting at Line 243 is too long and needs to be broken up into at least 3 paragraphs. 

There are a lot of citations of other people's work, but there are few discussions to how YOUR findings support these references. Perhaps more "Our current findings support ..."

There are also many general statements to why oral health and systemic health is important, but very little discussion on why your research is contributing to solving the problems. 

Response:

Discussion:

Line 243: We acknowledge your feedback on the Discussion section, and we agree that it requires improvement. We have taken your advice and restructured the paragraph starting at Line 243 into three separate paragraphs to improve readability and flow. By clearly pointing out the connections between our findings and the existing literature, we aim to strengthen the significance of our study in the broader context of oral health and systemic diseases. Additionally, we have included more explicit statements like "Our current findings support..." to highlight the relevance of our research in contributing to solving the issues related to oral health and systemic health.

Comments on the Quality of English Language

Line 47: "remarkable and invaluable treasure" is odd

Please clarify which "disease" you are referring to (systemic diseases, oral diseases, periodontal diseases)

Response:

Line 47: We appreciate your input regarding the phrase "remarkable and invaluable treasure." We have replaced this with a more appropriate expression: “A healthy mouth is intricately linked with overall health and the quality of one's life.”

We have made sure to specify which "disease"(periodontal disease) is being referred to throughout the manuscript. This will prevent any confusion and help readers to better understand our study's focus.

We hope that these revisions adequately address your concerns and improve the overall clarity and impact of our manuscript. Your constructive feedback has been instrumental in enhancing the quality of our work, and we are grateful for the opportunity to make these improvements.

Once again, we thank you for your thorough review and invaluable comments.

Sincerely,

Ramona Dumitrescu

Translational and Experimental Clinical Research Centre in Oral Health, Department of Preventive, Community Dentistry and Oral Health, University of Medicine and Pharmacy “Victor Babes”, 300040 Timisoara, Romania

[email protected]

Tel: +40745505481

Round 2

Reviewer 2 Report

Much better presentation of the scientific results in the second version of your manuscript, with more important literature and less reiterations in the discussion part.

Author Response

Much better presentation of the scientific results in the second version of your manuscript, with more important literature and less reiterations in the discussion part.

Response:

Dear Reviewer,

Thank you for your feedback on the second version of the manuscript. We greatly appreciate your observation regarding the improved presentation of the scientific results. We've worked to incorporate more pertinent literature and minimize reiterations in the discussion section. Your insights have been invaluable in enhancing the quality of our work.

Sincerely,

Ramona Dumitrescu

Translational and Experimental Clinical Research Centre in Oral Health, Department of Preventive, Community Dentistry and Oral Health, University of Medicine and Pharmacy “Victor Babes”, 300040 Timisoara, Romania

[email protected]  Tel: +40745505481